# Intra-Processing Methods for Debiasing Neural Networks

**Yash Savani**
Abacus.AI
San Francisco, CA 94103
yash@abacus.ai

**Colin White**
Abacus.AI
San Francisco, CA 94103
colin@abacus.ai

**Naveen Sundar Govindarajulu**
RAIR Lab RPI
Troy, NY 12180
naveensundarg@gmail.com

## Abstract

As deep learning models become tasked with more and more decisions that impact human lives, such as criminal recidivism, loan repayment, and face recognition for law enforcement, bias is becoming a growing concern. Debiasing algorithms are typically split into three paradigms: pre-processing, in-processing, and post-processing. However, in computer vision or natural language applications, it is common to start with a large generic model and then fine-tune to a specific use-case. Pre- or in-processing methods would require retraining the entire model from scratch, while post-processing methods only have black-box access to the model, so they do not leverage the weights of the trained model. Creating debiasing algorithms specifically for this fine-tuning use-case has largely been neglected.

In this work, we initiate the study of a new paradigm in debiasing research, *intra-processing*, which sits between in-processing and post-processing methods. Intra-processing methods are designed specifically to debias large models which have been trained on a generic dataset, and fine-tuned on a more specific task. We show how to repurpose existing in-processing methods for this use-case, and we also propose three baseline algorithms: random perturbation, layerwise optimization, and adversarial debiasing. We evaluate these methods across three popular datasets from the AIF360 toolkit, as well as on the CelebA faces dataset. [1]

## 1 Introduction

The last decade has seen a huge increase in applications of machine learning in a wide variety of domains such as credit scoring, fraud detection, hiring decisions, criminal recidivism, loan repayment, face recognition, and so on [43, 7, 45, 3, 36]. The outcome of these algorithms are impacting the lives of people more than ever. There are clear advantages in the automation of classification tasks, as machines can quickly process thousands of datapoints with many features. However, algorithms are susceptible to bias towards individuals or groups of people from a variety of sources [49, 46, 47, 60, 8, 28, 59].

For example, facial recognition algorithms are currently being used by the US government to match application photos from people applying for visas and immigration benefits, to match mugshots, and to match photos as people cross the border into the USA [24]. However, recent studies showed that many of these algorithms exhibit bias based on race and gender [23]. For example, some of the algorithms were 10 or 100 times more likely to have false positives for Asian or Black people, compared to white people. When used for law enforcement, it means that a Black or Asian person is more likely to be arrested and detained for a crime they didn't commit [1].

Motivated by the discovery of biased models in real-life applications, the last few years has seen a huge growth in the area of fairness in machine learning. Dozens of formal definitions of fairness have been proposed [44], and many algorithmic techniques have been developed for debiasing according to these definitions [58]. Many debiasing algorithms fit into one of three categories: pre-processing, in-processing, or post-processing [14, 5]. Pre-processing techniques make changes to the data itself, in-processing techniques are methods for training machine learning models tailored to making fairer models, and post-processing techniques modify the final predictions outputted by a (biased) model.

However, as datasets become larger and training becomes more computationally intensive, especially in the case of computer vision and natural language processing, it is becoming increasingly more common in applications to start with a very large pretrained model, and then fine-tune for the specific use-case [50, 30, 10, 57]. In fact, PyTorch offers several pretrained models, all of which have been trained for dozens of GPU hours on ImageNet [51]. Pre-, in-, and post-processing debiasing methdos are of little help here: pre- and in-processing methods would require retraining the entire model from scratch, and post-processing methods would not make use of the full power of the model.

In this work, we initiate the study of *intra-processing* methods for debiasing neural networks. An intra-processing method is defined as an algorithm which has access to a trained model and a dataset (which typically differs from the original training dataset), and outputs a new model which gives debiased predictions on the target task (typically by updating or augmenting the weights of the original model). We propose three different intra-processing baseline algorithms, and we also show how to repurpose a popular in-processing algorithm [61] to the intra-processing setting. All of the algorithms we study work for any group fairness measure and any objective which trades off accuracy with bias. Our first baseline is a simple random perturbation algorithm, which iteratively adds multiplicative noise to the weights of the neural network and then picks the perturbation which maximizes the chosen objective. Our next baseline optimizes the weights of each layer using GBRT [20]. Finally, we propose adversarial methods for fine-tuning. Adversarial training was recently used as an in-processing method for debiasing [61], by training a critic model to predict the protected attribute of datapoints, to ensure that the predictions are not correlated with the protected attribute. We modify this approach to be an intra-processing, and we also propose a new, more direct approach which trains a critic to *directly measure the bias* of the model weights, which gives us a differentiable proxy for bias, enabling the use of gradient descent for debiasing.

We compare the above techniques with one in-processing algorithm and three post-processing algorithms from prior work: adversarial debiasing [61], reject option classification [31], equalized odds post-processing [26], and calibrated equalized odds post-processing [52]. We run experiments with three fairness datasets from AIF360 [5], as well as the CelebA dataset [39], with three popular fairness definitions. We show that intra-processing is much more effective than post-processing for the fine-tuning use case. We also show that the difficulty of post-hoc debiasing is highly dependent on the initial conditions of the original model. In particular, given a neural network trained to optimize accuracy, the variance in the amount of bias of the trained model is much higher than the variance in the accuracy, with respect to the random seed used for initializing the weights of the original model. Fairness research (and machine learning research as a whole) has seen a huge increase in popularity, and recent papers have highlighted the need for fair and reproducible results [55, 5]. To facilitate best practices, we run our experiments on the AIF360 toolkit [5] and open source all of our code.

**Our contributions.**    We summarize our main contributions below.

- We initiate the study of intra-processing algorithms for debiasing ML models. This framework sits in between in-processing and post-processing, and is realistic for many fine-tuning use cases.
- We study the nature of intra-processing techniques for debiasing neural networks, showing that the problem is sensitive to the initial conditions of the original model.
- We propose three baseline intra-processing algorithms, and we show how to repurpose popular in-processing algorithms into the intra-processing setting. We compare all algorithms across a variety of group fairness constraints and datasets.

## 2   Related Work

**Debiasing overview.**    There is a surging body of research on bias and fairness in machine learning. There are dozens of types of bias that can arise [41], and dozens of formal definitions of fairness

have been proposed [44]. Popular definitions include statistical parity/demographic parity [17, 34], equal opportunity (a subset of equalized odds) [25], and average absolute odds [5]. For an overview of fairness definitions and techniques, see [5, 58]. Recently, the AIF360 toolkit was established to facilitate best practices in debiasing experiments [5]. Recently, a meta-algorithm was developed for in-processing debiasing by reducing fairness measures to convex optimization problems [9]. Another work treats debiasing as an empirical risk minimization problem [15]. Yet another work adds the fairness constraints as regularizers in the machine learning models [6]. There is also prior work using adversarial learning to debias algorithms [61]. To the best of our knowledge, no prior work has designed an intra-processing algorithm using adversarial learning.

**Post-processing methods.** Many post-processing debiasing algorithms have been proposed, which all assume black-box access to a biased classifier [32, 39, 25, 52, 31]. We describe a few of these techniques in Section 5. Currently, most of these techniques have only been established for specific fairness measures. In natural language processing, there is very recent work which decreases the bias present in pretrained models such as BERT or ELMo [38], however, these techniques were not shown to work on tabular or computer vision datasets.

**Debiasing in computer vision.** A recent work studied methods for debiasing CelebA with respect to gender [59]. However, the study was limited to in-processing techniques focusing on achieving the highest accuracy in light of distribution shift with respect to correlations present in the data. There has also been work in measuring and mitigating bias in gender classfication in computer vision [8].

## 3 Preliminaries

In this section, we give notation and definitions used throughout the paper. Given a dataset split into three parts, $\mathcal{D}_{\text{train}}$, $\mathcal{D}_{\text{valid}}$, $\mathcal{D}_{\text{test}}$, let $(\boldsymbol{x}_i, Y_i)$ denote one datapoint, where $\boldsymbol{x}_i \in \mathbb{R}^d$ contains $d$ features including one binary protected attribute $A$ (e.g., identifying as female or not identifying as female), and $Y_i \in \{0, 1\}$ is the label. Denote the value of the protected feature for $\boldsymbol{x}_i$ as $a_i$. We denote a trained neural network by a function $f_\theta : \mathbb{R}^d \to [0, 1]$, where $\theta$ denotes the trained weights. We often denote $f_\theta(\boldsymbol{x}_i) = \hat{Y}_i$, the output predicted probability for datapoint $\boldsymbol{x}_i$. Finally, we refer to a set of labels in a dataset $\mathcal{D}$ as $\mathcal{Y}$.

**Fairness measures.** We now give an overview of group fairness measures used in this work. Given a dataset $\mathcal{D}$ with labels $\mathcal{Y}$, protected attribute $A$, and a set of predictions $\hat{\mathcal{Y}} = \{f_\theta(\boldsymbol{x}) \mid \boldsymbol{x} \in \mathcal{D}\}$ from some neural network $f_\theta$, we define the true positive and false positive rates as

$$TPR_{A=a}(\mathcal{D}, \hat{\mathcal{Y}}) = \frac{\left|\{i \mid \hat{Y}_i = Y_i = 1, a_i = a\}\right|}{\left|\{i \mid \hat{Y}_i = Y_i = 1\}\right|} = P_{(\boldsymbol{x}_i, Y_i) \in \mathcal{D}}(\hat{Y}_i = 1 \mid a_i = a, Y_i = 1),$$

$$FPR_{A=a}(\mathcal{D}, \hat{\mathcal{Y}}) = \frac{\left|\{i \mid \hat{Y}_i = 1, Y_i = 0, a_i = a\}\right|}{\left|\{i \mid \hat{Y}_i = 1, Y_i = 0\}\right|} = P_{(\boldsymbol{x}_i, Y_i) \in \mathcal{D}}(\hat{Y}_i = 1 \mid a_i = a, Y_i = 0),$$

*Statistical Parity Difference (SPD)*, or demographic parity difference [17, 34], measures the difference in the probability of a positive outcome between the protected and unprotected groups. Formally,

$$SPD(\mathcal{D}, \hat{\mathcal{Y}}, A) = P_{(\boldsymbol{x}_i, Y_i) \in \mathcal{D}}(\hat{Y}_i = 1 \mid a_i = 0) - P_{(\boldsymbol{x}_i, Y_i) \in \mathcal{D}}(\hat{Y}_i = 1 \mid a_i = 1).$$

*Equal opportunity difference (EOD)* [25] measures the difference in TPR for the protected and unprotected groups. Equal opportunity is identical to *equalized odds* in the case where the protected feature and labels are binary. Formally, we have

$$EOD(\mathcal{D}, \hat{\mathcal{Y}}, A) = TPR_{A=0}(\mathcal{D}, \hat{\mathcal{Y}}) - TPR_{A=1}(\mathcal{D}, \hat{\mathcal{Y}}).$$

*Average Odds Difference (AOD)* [5] is defined as the average of the difference in the false positive rates and true positive rates for unprivileged and privileged groups. Formally,

$$AOD(\mathcal{D}, \hat{\mathcal{Y}}, A) = \frac{(FPR_{A=0}(\mathcal{D}, \hat{\mathcal{Y}}) - FPR_{A=1}(\mathcal{D}, \hat{\mathcal{Y}})) + (TPR_{A=0}(\mathcal{D}, \hat{\mathcal{Y}}) - TPR_{A=1}(\mathcal{D}, \hat{\mathcal{Y}}))}{2}.$$

**Optimization techniques.** Zeroth order (non-differentiable) optimization is used when the objective function is not differentiable (as is the case for most definitions of group fairness). This is also called black-box optimization. Given an input space $W$ and an objective function $\mu$, zeroth order optimization seeks to compute $w^* = \arg\min_{w \in W} \mu(w)$. Leading methods for zeroth order optimization when function queries are expensive (such as optimizing a deep network) include gradient-boosted regression trees (GBRT) [20, 40] and Bayesian optimization (BO) [53, 19, 56], however BO struggles with high-dimensional data. In contrast, first-order optimization is used when it is possible to take the derivative of the objective function. Gradient descent is an example of a first-order optimization technique.

## 4 Methodology

In this section, we describe three new intra-processing algorithms for debiasing neural networks. First we give more notation and formally define the different types of debiasing algorithms.

Given a neural network $f_\theta$, we sometimes drop the subscript $\theta$ when it is clear from context. We denote the last layer of $f$ by $f^{(\ell)}$, and we assume that $f = f^{(\ell)} \circ f'$, where $f'$ is all but the last layer of the neural network. Our layer-wise optimization algorithm assumes that $f$ is feed-forward, that is, $f = f^{(\ell)} \circ \cdots \circ f^{(1)}$ for functions $f^{(1)}, f^{(2)}, \ldots, f^{(\ell)}$. The performance of the model is given by a performance measure $\rho$. For a set of data points $\mathcal{D}$, given the set of true labels $\mathcal{Y}$ and the set of predicted labels $\hat{\mathcal{Y}} = \{f(\boldsymbol{x}_i) \mid (\boldsymbol{x}_i, Y_i) \in \mathcal{D}'\}$, the performance is $\rho(\mathcal{Y}, \hat{\mathcal{Y}}) \in [0, 1]$. Common performance measures include accuracy, precision, recall, or AUC ROC (area under the ROC curve). In our experiments we use balanced accuracy as our performance measure. We also define a bias measure $\mu$, given as $\mu(\mathcal{D}, \hat{\mathcal{Y}}, A) \in [0, 1]$, such as one defined in Section 3.

The goal of any debiasing algorithm is to decrease the bias $\mu$, without sacrificing performance $\rho$ too much. Many prior works have observed that fairness comes at the price of accuracy for many datasets, even when using large models such as deep networks [5, 58, 11]. Therefore, a common technique is to maximize the performance subject to a constraint on the bias, e.g., $\mu < 0.05$. Concretely, we define an objective function as follows.

$$\phi_{\mu,\rho,\epsilon}(\mathcal{D}, \hat{\mathcal{Y}}, A) = \begin{cases} \rho \text{ if } \mu < \epsilon, \\ 0 \text{ otherwise.} \end{cases} \tag{1}$$

An in-processing debiasing algorithm takes as input the training and validation datasets and outputs a model $f$ which seeks to maximize $\phi_{\mu,\rho,\epsilon}$. An intra-processing algorithm takes in the validation dataset and a trained model $f$ with weights $\theta$ (typically $f$ was trained to optimize the performance $\rho$), and outputs fine-tuned weights $\theta'$ such that $f_{\theta'}$ maximizes the objective $\phi_{\mu,\rho,\epsilon}$. A post-processing debiasing algorithm takes as input the validation dataset as well as a set of predictions $\hat{\mathcal{Y}}$ on the validation dataset (typically coming from a model $f$ which was optimized for $\rho$), and outputs a post-processing function $h : [0, 1] \to \{0, 1\}$ which performs post-processing on predictions so that the final predictions optimize $\phi_{\mu,\rho,\epsilon}$. Note that intra-processing and post-processing debiasing algorithms are useful in different settings. Post-processing algorithms are useful when there is no access to the original model. Intra-processing algorithms are useful when there is access to the original model, or when the prediction is over a continuous feature. Now we present three new intra-processing techniques.

**Random perturbation.** Our first algorithm is a simple iterative random procedure, *random perturbation*. In every iteration, each weight in the neural network is multiplied by a Gaussian random variable with mean 1 and standard deviation 0.1. In case the model $f$ outputs probabilities, we find the threshold $\tau$ such that $\hat{\mathcal{Y}}_\tau = \{\mathbb{I}\{\hat{Y} > \tau\}\}_{\hat{Y} \in \hat{\mathcal{y}}}$ maximizes $\phi_{\mu,\rho,\epsilon}(\mathcal{Y}, \hat{\mathcal{Y}}_\tau, A)$. We run $T$ iterations and output the perturbed weights which maximize $\phi_{\mu,\rho,\epsilon}$ on the validation set. See Algorithm 1. We show in the next section that despite its simplicity, this model performs well on many datasets and

fairness measures, and therefore we recommend this algorithm as a baseline in future intra-processing debiasing applications. A natural follow-up question is whether we can do even better by using an optimization algorithm instead of random search. This is the motivation for our next approach.

---

**Algorithm 1** Random Perturbation

---

1: **Input:** Trained model $f$ with weights $\theta$, validation dataset $\mathcal{D}_{\text{valid}}$, objective $\phi_{\mu,\rho,\epsilon}$, parameter $T$
2: Set $\theta^* = \theta$, val$^* = -\infty$, and $\tau^* = 0$
3: **for** $i = 1$ to $T$ **do**
4:     Sample $q_j \sim \mathcal{N}(1, 0.1)$ for all $j \in \{1, 2, ..., |\theta|\}$
5:     $\theta'_j = \theta_j \cdot q_j$
6:     Select threshold $\tau \in [0, 1]$ which maximizes the objective $\phi_{\mu,\rho,\epsilon}$ on the validation set
7:     Set val $= \phi_{\mu,\rho,\epsilon}(\mathcal{D}_{\text{valid}}, \{\mathbb{I}\{f_{\theta'}(\boldsymbol{x}) > \tau\} \mid (\boldsymbol{x}, Y) \in \mathcal{D}_{\text{valid}}\}, A)$
8:     If val $>$ val$^*$, set val$^* = $ val, $\theta^* = \theta'$, and $\tau^* = \tau$.
9: **end for**
10: **Output:** $\theta^*$, $\tau^*$

---

**Layer-wise optimization.** Our next method fine-tunes the model by debiasing individual layers using zeroth order optimization. Intuitively, an optimization procedure will be much more effective than random perturbations, but it is computationally expensive and does not scale as well, so we can only run optimization on individual layers. Given a model, assume the model can be decomposed into several functions $f = f^{(\ell)} \circ \cdots \circ f^{(1)}$ For example, a feed-forward neural network with $\ell$ layers can be decomposed in this way. We denote the trained weights of each component by $\theta_1, \ldots, \theta_\ell$, respectively. Now assume that we have access to a zeroth order optimizer $\mathcal{A}$, which takes as input a model $f = f^{(\ell)} \circ \cdots \circ f^{(1)}$, weights $\theta = (\theta_1, \ldots, \theta_\ell)$, dataset $\mathcal{D}_{\text{valid}}$, and an index $i$. The optimizer returns weights $\theta'_i$, optimized with respect to to $\phi_{\mu,\rho}$. In Algorithm 2, we set the optimizer to be gradient-boosted regression trees (GBRT) [20, 40], a leading technique for black box optimization which converts shallow regression trees into strong learners. GBRT iteratively constructs a posterior predictive model using the weights to make predictions and uncertainty estimates for each potential set of weights $\theta'$. To trade off exploration and exploitation, the next set of weights to try is chosen using lower confidence bounds (LCB), a popular acquisition function (e.g., [29]). Formally, $\phi_{\text{LCB}}(\theta') = \hat{\theta}' - \beta\hat{\sigma}$, in which we assume our model's posterior predictive density follows a normal distribution with mean $\hat{\theta}'$ and standard deviation $\hat{\sigma}$. $\beta$ is a tradeoff parameter that can be tuned. See Algorithm 2. Note that this algorithm can be easily generalized to optimize multiple layers at once, but this comes at the price of runtime. For example, running GBRT on the entire neural network would be more powerful than the random permutation algorithm but is prohibitively expensive.

---

**Algorithm 2** Layer-wise optimization

---

1: **Input:** Trained model $f = f^{(\ell)} \circ \ldots \circ f^{(1)}$ with weights $\theta_1, \ldots, \theta_\ell$, objective $\phi_{\mu,\rho,\epsilon}$, black-box optimizer $\mathcal{A}$
2: Set $\theta^* = \emptyset$, val$^* = -\infty$, and $\tau^* = 0$
3: **for** $i = 1$ to $\ell$ **do**
4:     Run optimizer $\mathcal{A}$ to optimize weights $\theta_i$ to $\theta'_i$ with respect to $\phi_{\mu,\rho,\epsilon}$.
5:     Select threshold $\tau \in [0, 1]$ which maximizes objective $\phi_{\mu,\rho,\epsilon}$
6:     Set val $= \phi_{\mu,\rho,\epsilon}(\mathcal{D}_{\text{valid}}, \{\mathbb{I}\{f_{\theta'}(\boldsymbol{x}) > \tau\} \mid (\boldsymbol{x}, Y) \in \mathcal{D}_{\text{valid}}\}, A)$, where $\theta' = (\theta_1, ..., \theta'_i, ..., \theta_\ell)$
7:     If val $>$ val$^*$ set val$^* = $ val, $\theta^* = \theta'$, and $\tau^* = \tau$.
8: **end for**
9: **Output:** $\theta^*$, $\tau^*$

---

**Adversarial fine-tuning.** The previous two methods rely on zeroth order optimization techniques because most group fairness measures such as statistical parity difference and equalized odds are non-differentiable. Our last technique casts the problem of debiasing as first-order optimization by using adversarial learning. The idea behind the adversarial method is that we train a critic model to predict the amount of bias in a minibatch. We sample the datapoints in a minibatch randomly and with replacement. This statistical bootstrapping approach to creating a minibatch means that if the critic can predict the bias in a minibatch accurately, then it can predict the bias in the model with

respect to the validation set reasonably well. Therefore, the critic effectively acts as a differentiable proxy for bias, which makes it possible to debias the original model using gradient descent.

The adversarial algorithm works by alternately iterating between training the critic model $g$ using the predictions from $f$, and fine-tuning the predictive model $f$ with respect to a custom function designed to be differentiable while still maximizing the non-differentiable objective function $\phi_{\mu,\rho,\epsilon}$ using the bias proxy $\hat{\mu}$ from $g$. The custom function we use to emulate the objective function while still being differentiable is given by $\max\{1, \lambda \cdot (|\hat{\mu}| - \epsilon + \delta) + 1\} \cdot Loss(y, \hat{y})$ where $\hat{y}$ is the predicted label, $y$ is the real label, $\lambda, \delta$ are hyperparameters, and Loss is a generic loss function such as binary cross-entropy. This function multiplies the task specific loss by a coefficient that is 1 if the absolute bias is less than $\epsilon - \delta$, otherwise the coefficient is $\lambda \cdot |\hat{\mu}|$. Intuitively, this custom function optimizes the task specific loss subject to the absolute bias less than $\epsilon - \delta$. The hyperparameter $\lambda$ describes how strict the bias constraint should be. Note that the $g$ concatenates the examples in the minibatch and returns a single number that estimates the bias of the minibatch as the final output. See Algorithm 3. Note that BCELoss denotes the standard binary cross-entropy loss.

---

**Algorithm 3** Adversarial Fine-Tuning

---

1: **Input:** Trained model $f = f^{(\ell)} \circ f'$ with weights $\theta$, validation dataset $\mathcal{D}_{\text{valid}}$, parameters $\lambda, \epsilon, \delta, n, m, m', T$.
2: Set $g$ as the critic model with weights $\psi$.
3: **for** $i = 0$ to $n$ **do**
4:      **for** $j = 0$ to $m$ **do**
5:          Sample a minibatch $(\boldsymbol{X}_k, \boldsymbol{\mathcal{Y}}_k)$ with replacement from $\mathcal{D}_{\text{valid}}$
6:          Evaluate the bias in the minibatch, $\bar{\mu} \leftarrow \mu((\boldsymbol{X}_k, \boldsymbol{\mathcal{Y}}_k), f(\boldsymbol{X}_k))$.
7:          Update the critic model $g$ by updating its stochastic gradient

$$\nabla_\psi (\bar{\mu} - (g \circ f')(\boldsymbol{X}_k))^2$$

8:      **end for**
9:      **for** $j = 0$ to $m'$ **do**
10:        Sample a minibatch $(\boldsymbol{X}_k, \boldsymbol{\mathcal{Y}}_k)$ with replacement from $\mathcal{D}_{\text{valid}}$
11:        Update the original model by updating its stochastic gradient

$$\nabla_\theta \left[ \max\{1, \lambda \cdot (|(g \circ f')(\boldsymbol{X}_k)| - \epsilon + \delta) + 1\} \cdot \text{BCELoss}(\boldsymbol{\mathcal{Y}}_k, f(\boldsymbol{X}_k)) \right]$$

12:      **end for**
13:      Select threshold $\tau \in [0, 1]$ that minimizes the objective $\phi_{\mu,\rho}$
14: **end for**
15: **Output:** Debiased model $f$, threshold $\tau$

---

**Converting in-processing into intra-processing.** Since both in-processing and intra-processing algorithms optimize the weights of the neural network while training, it may be possible to convert existing in-processing algorithms into intra-processing algorithms. However, the in-processing algorithm needs to be able to run on a generic neural architecture, and there cannot be any specific weight-initialization step (since it is given a pretrained model as the starting point). Furthermore, the hyperparameters of the in-processing algorithm may need to be modified to better fit the fine-tuning paradigm. For example, the learning rate should be lowered, and the optimizer's influence on the earlier layers that are unlikely to contribute as much to the final result should be limited.

As an instructive example, we modify a popular in-processing fairness algorithm [61] to convert it to the intra-processing setting. This algorithm relies on a similar adversarial paradigm to our adversarial fine-tuning algorithm. The fundamental difference between the algorithm of [61] and our Algorithm 3 is that their algorithm uses the critic to predict the protected attribute and not to directly predict the bias for the minibatch. As a result, we modify our adversarial fine-tuning algorithm so the critic predicts the protected attribute, and we change the custom function to the objective provided by the original work. Finally we modify the hyperparameters so that they are better suited to the fine-tuning use case. For instance, we use a lower learning rate when fine-tuning the model.

# 5 Experiments

In this section, we experimentally evaluate the techniques laid out in Section 4 compared to baselines, on three datasets and with multiple fairness measures. To promote reproducibility, we release our code at `https://github.com/abacusai/intraprocessing_debiasing` and we use datasets from the AIF360 toolkit [5] and a popular image dataset. Each dataset contains one or more binary protected features(s) and a binary label. We briefly describe them below.

The COMPAS dataset is a commonly used dataset in fairness research, consisting of over 10,000 defendants with 402 features [18]. The goal is to predict the recidivism likelihood for an individual [2]. We run separate experiments using *race* and also *gender* as protected attributes. The Adult Census Income (ACI) dataset is a binary classification dataset from the 1994 USA Census bureau database in which the goal is to predict whether a person earns above $50,000 [16]. There are over 40,000 data points with 15 features. We use *gender* and *race* as the protected attribute. The Bank Marketing (BM) dataset is from the phone marketing campaign of a Portuguese bank. There are over 48,000 datapoints consisting of 17 categorical and quantitative features. The goal is to predict whether a customer will subscribe to a product [42]. The protected feature is whether or not the customer is older than 25. The CelebA dataset [39] is a popular image dataset used in computer science research. This dataset consists of over 200000 images of celebrity head-shots, along with attributes such as "smiling", "young", and "gender". A number of groups have trained neural networks to predict these labels simply by looking at the image. Machine learning researchers/practitioners should strive to get rid of datasets that give binary labels to attributes such as gender and attractiveness [13], but our techniques will help to debias models that have already been trained on this type of dataset. Since the CelebA dataset contains the identity of each headshot, we can also debias the model with respect to race. We give the full details of how we used the CelebA dataset in the full version of this paper.

**Bias sensitivity to initial model conditions.** First, we run experiments to compute the amount of variance in the bias scores of the initial models. Neural networks have a huge number of local minima. Hyperparameters such as the optimizer and learning rate, and even the initial random seed, cause the model to converge to different local minima [37]. Techniques such as the Adam optimizer and early stopping with patience have been designed to allow neural networks to consistently reach local minima with high accuracies [33, 21]. However, there is no guarantee on the amount of bias. In particular, the local minima found by neural networks may have large differences in the amount of bias, and therefore, there may be very high variance on the amount of bias exhibited by neural networks just because of the random seed. Every local optima has a different set of weights. If the weights of the model at a specific local optimum rely heavily on the protected feature, removing the bias from such a model by updating the weights is harder than removing the bias from a model whose weights do not rely on the protected feature as heavily. We compute the mean and the standard deviation of three fairness measures, as well as accuracy, for a neural network trained with 10 different initial random seeds, across three datasets. We see that the standard deviation of the bias score is an order of magnitude higher than the standard deviation of the accuracy. See the full version of the paper for the full results. We also plot the contribution of each individual weight to the bias score, for a neural network. We show that the contribution of the weights to the bias score are sensitive to the initial random seed.

## 5.1 Intra-processing debiasing experiments

Now we present our main experimental study by comparing four intra-processing and three post-processing debiasing methods across four datasets and three fairness measures. This includes one in-processing algorithm that we have adapted to the intra-processing setting. First we briefly describe the baseline post-processing algorithms that we tested.

The *reject option classification* post-processing algorithm [31] defines a critical region of points in the protected group whose predicted probability is near $0.5$, and flips these labels. This algorithm is designed to minimize statistical parity difference. The *equalized odds* post-processing algorithm [25] defines a convex hull based on the bias rates of different groups, and then flips the label of data points that fall inside the convex hull. This algorithm is designed to minimize equal opportunity difference. The *Calibrated equalized odds* post-processing algorithm [52] defines a base rate of bias for each group, and then adds randomness based on the group into the classifier until the bias rates converge.

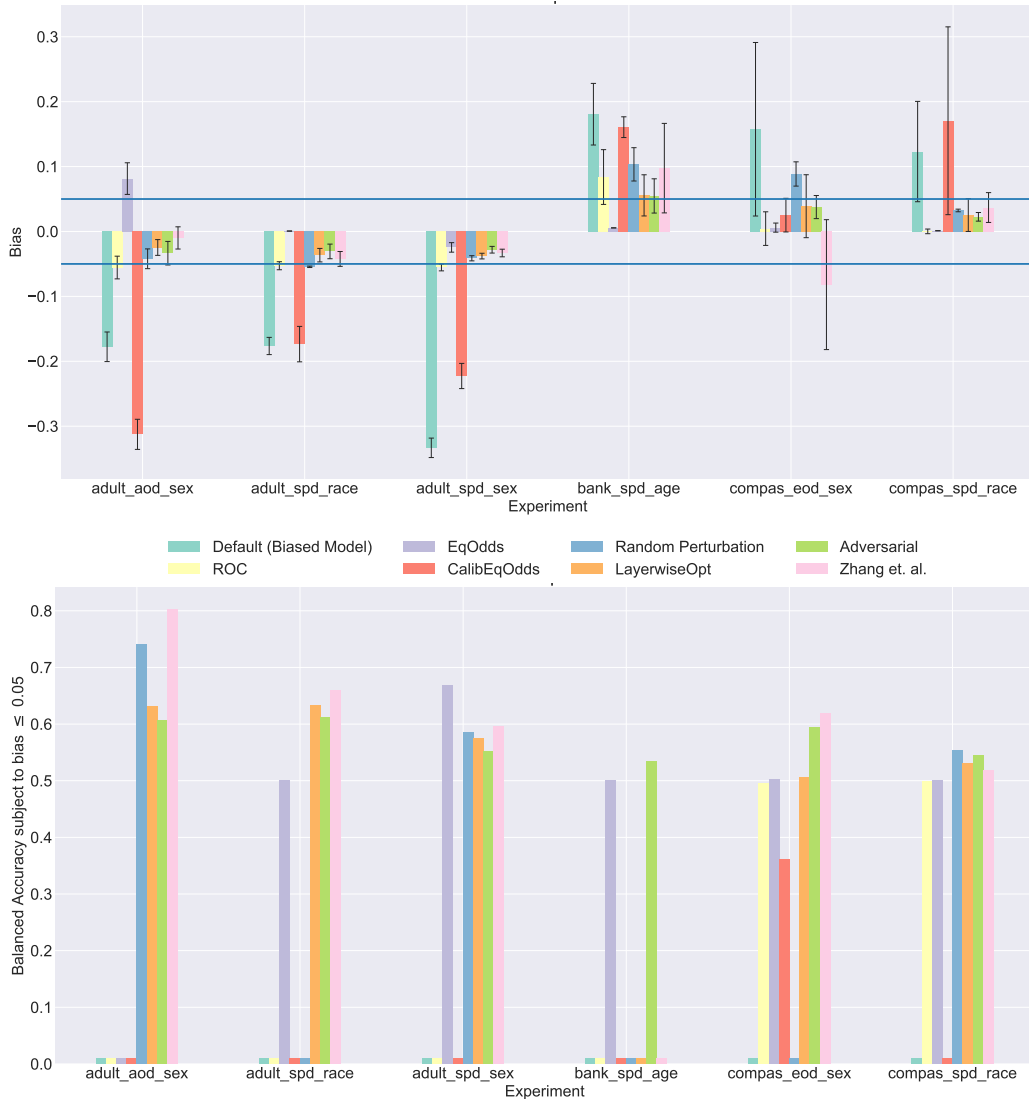

Figure 1: Results for tabular datasets over 5 runs with different seeds. mean Bias with std error bars (Top) and the median of the objective function in Equation 1 (Bottom).

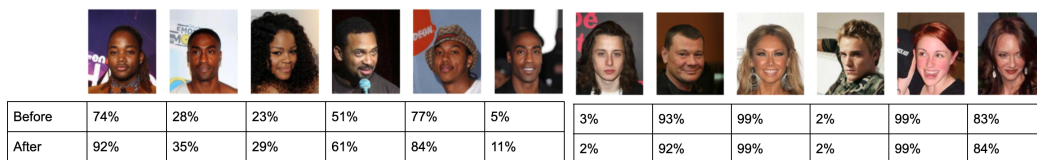

| | | | | | | | | | | | |
|---|---|---|---|---|---|---|---|---|---|---|---|
| Before | 74% | 28% | 23% | 51% | 77% | 5% | 3% | 93% | 99% | 2% | 99% | 83% |
| After | 92% | 35% | 29% | 61% | 84% | 11% | 2% | 92% | 99% | 2% | 99% | 84% |

Figure 2: Probability of smiling on the CelebA dataset, before and after debiasing w.r.t. race.

This algorithm is designed to minimize equal opportunity difference. For these algorithms, we use the implementations in the AIF360 repository [5].

Now we explain the experimental setup for the tabular datasets. Our initial model consists of a feed-forward neural network with 10 fully-connected layers of size 32, with a BatchNorm layer between each fully-connected layer, and a dropout fraction of 0.2. The model is trained with the Adam optimizer and an early-stopping patience of 100 epochs. The loss function is the binary cross-entropy loss. We use the validation data as the input for the intra-processing methods, with the objective function set to Equation 1 with $\epsilon = 0.05$. We modified the hyperparameters so that

Table 1: Results on the CelebA datasets for a pretrained ResNet with three initial random seeds. Results are the balanced accuracy scores after fine-tuning. The crossed out scores are those that did not have biases lower than 0.05.

| | Default | ROC | EqOdds | CalibEqOdds | Random | LayerwiseOpt | Adversarial |
|---|---|---|---|---|---|---|---|
| 1 | ~~0.533~~ | ~~0.533~~ | ~~0.983~~ | 0.519 | ~~0.567~~ | ~~0.530~~ | 0.914 |
| 2 | ~~0.523~~ | 0.521 | ~~0.983~~ | 0.487 | 0.529 | 0.508 | 0.917 |
| 3 | 0.535 | 0.533 | ~~0.982~~ | 0.514 | ~~0.591~~ | 0.529 | 0.905 |

each method took roughly 30 minutes. We run each algorithm on 5 neural networks initialized with different random seeds and aggregate the results. We publish the median objective scores for the tabular datasets as the mean would not accurately portray the expected results since some runs may return an objective score of 0.0. We also publish the mean and std error bars for the bias scores. We limit the plots to those for which the default biased algorithm did not achieve a positive objective score. See Figure 1.

Finally, we run experiments on the CelebA dataset. We use a ResNet [27] pretrained on ImageNet from the PyTorch library [51] as our initial model. We publish the balanced accuracy scores of the algorithms when trained on a binary classification task to identify whether the celebrity in the image is classified as smiling or not. See Figure 1. We also publish a series of images along with the predicted probability that the celebrity is smiling. These are from the adversarial fine-tuning model. See Figure 2. For the image dataset we consider the default biased model, the 3 post-processing algorithm, and the 3 novel intra-processing algorithms.

**Discussion.** We see that the intra-processing methods significantly outperform the post-processing methods, sometimes even on the fairness metric for which the post-processing method was designed. We note that there are two caveats. First, the three intra-processing methods had access to the objective function in Equation 1, while the post-processing methods are only designed to minimize their respective fairness measures. However, as seen in Figure 1, sometimes the intra-processing methods simultaneously achieve higher objective scores and lower bias compared to the post-processing methods, making the intra-processing methods dominate them pareto-optimally. Second, intra-processing methods are more powerful than post-processing methods, since post-processing methods do not modify the weights of the original model. Post-processing methods are more appropriate when the model weights are unavailable or when computation time is constrained, and intra-processing methods are more appropriate when higher performance is desired. We find that all the intra-processing algorithms tend to do well on the tabular datasets. However making sure that their bias scores remain below the threshold on the test is not as easy as there were not enough rows. Using regularization techniques helped to ensure that the scores remained consistent even over the test set. We find that when the dataset and the model become more complex as is the case with the CelebA dataset and the ResNet model the more complex algorithms like adversarial fine-tuning tend to perform better than the random perturbation and layerwise optimization algorithm. This indicates that when dealing with more complex datasets and models, using complex intra-processing models like adversarial fine-tuning may be a better fit for the problem.

# 6   Conclusion

In this work, we initiate the study of a new paradigm in debiasing research, *intra-processing*, which sits between in-processing and post-processing methods, and is designed for fairly fine-tuning large models. We define three new intra-processing algorithms: random perturbation, adversarial fine-tuning, and layer-wise optimization, and we repurpose a popular in-processing algorithm to work for intra-processing. In our experimental study, first we show that the amount of bias is sensitive to the initial conditions of the original neural network. Then we give an extensive comparison of four intra-processing methods and three post-processing methods across three tabular datasets, one image dataset, and three popular fairness measures. We show that the intra-processing algorithms outperform the post-processing methods.

# 7  Broader Impact

Deep learning algorithms are more prevalent than ever before. The technology is becoming more and more integrated into society, and is used in high-stakes applications such as criminal recidivism, loan repayment, and hiring decisions [43, 7, 45, 3]. It is also becoming increasingly more evident that many of these algorithms are biased from various sources [49, 46, 47]. Using technology for life-changing events which make prejudiced decisions will only deepen the divides that exist in society, and the need to address these issues is higher than ever [4, 48].

Our work seeks to decrease the negative effects that biased deep learning algorithms have on society. Intra-processing methods, which work for any group fairness measure, will be applicable to large existing deep learning models, since the networks need not be retrained from scratch. Furthermore, we present simple techniques (random perturbation) as well as more complex and strong techniques (adversarial fine-tuning). Since we study the nature of intra-processing debiasing and present a study comparing prior work to our algorithms, our work may facilitate future work in intra-processing debiasing techniques.

**Impact on bias in judicial applications**  We briefly discuss how intra-processing methods for debiasing could help in judicial settings. Some machine learning algorithms which are prejudiced have been used in judicial applications in the past [54, 22]. Studies and investigations have found that many of the algorithms have some form of bias [35]. Moreover, different entities using the same model might prefer to use different fairness measures and some of these measures might be incompatible [12]. Generally, the entities that build and use the applications are not the same. Therefore, due to legal and licensing issues, the entity using the application may not have access to the training dataset. This precludes the use of pre-processing and in-processing methods for debiasing. The entity using the model usually has its own dataset available (e.g. a local court tracking their recidivism rates). This makes intra-processing and post-processing techniques the only viable methods for debiasing.

# 8  Acknowledgments

We thank the anonymous reviewers for their helpful suggestions. This work was completed while NSG was working at Abacus.AI.

## Footnotes

[1]See the full-length paper here: https://arxiv.org/abs/2006.08564. Our code is available at https://github.com/abacusai/intraprocessing_debiasing.

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
