[Supplementary Material 1]

# A Study on Post-Hoc Methods for Debiasing Neural Networks

## Abstract

As deep learning models become tasked with more and more decisions that impact human lives, such as hiring, criminal recidivism, and loan repayment, bias is becoming a growing concern. This has led to dozens of definitions of fairness, and numerous algorithmic techniques to improve the fairness of neural networks. Most debiasing algorithms require retraining a neural network from scratch, however, this is not feasible in many applications, especially when the model takes days to train or when the full training dataset is no longer available.

In this work, we present the first study on post-hoc methods for debiasing neural networks. First we study the nature of the problem, showing that the difficulty of post-hoc debiasing is highly dependent on the initial conditions of the original model. Then we define three new fine-tuning techniques: random perturbation, layer-wise optimization, and adversarial fine-tuning. All three techniques work for any group fairness constraint. We give a comparison with six algorithms - three popular post-processing debiasing algorithms and our three proposed methods - across three datasets and three popular bias measures. We show that no post-hoc debiasing technique dominates all others, and we identify settings in which each algorithm performs the best.

## 1 Introduction

The last decade has seen a huge increase in applications of machine learning in a wide variety of domains such as credit scoring, fraud detection, hiring decisions, criminal recidivism, loan repayment, and so on [36, 6, 39, 2]. The outcome of these algorithms are impacting the lives of people more than ever. There are clear advantages in the automation of classification tasks, as machines can quickly process thousands of datapoints with many features. However, algorithms are susceptible to bias towards individuals or groups of people from a variety of sources [43, 40, 41]. For example, Correctional Offender Management Profiling for Alternative Sanctions (COMPAS) is a computer software which determines the risk of a defendant committing a future crime. United States judges consult the software to decide whether or not a defendant should be granted bail or pretrial release. It was found that this software is biased against African-Americans [17]. Due to the current social climate in many countries, the need to address these issues is higher than ever [3, 42].

Motivated by these findings, the last few years has seen a huge growth in the area of fairness in machine learning. Dozens of formal definitions of fairness have been proposed [38], and many algorithmic techniques have been developed for debiasing according to these definitions [49]. While substantial progress has been made, the majority of techniques have been developed as pre-processing or in-processing methods. In other words, most techniques are developed to run before or during the machine learning model is trained, either as an add-on, or as a newly proposed algorithm. Only a handful of debiasing methods run after the training has been completed, either as fine-tuning methods

or post-processing methods [4]. However, as datasets become larger and training becomes more computationally intensive, especially in the case of neural networks, there is a growing need for debiasing algorithms which do not require retraining a model from scratch. Additionally, some applications may require debiasing an existing model without full access to the training dataset, due to regulatory requirements or privacy concerns. For example, this may be true any time the entities which build the model are different from the entities which deploy the model. Furthermore, every previously proposed post-processing method has been designed for its own fairness measure. Due to the diversity in fairness definitions, and since reduction of more than one fairness measure may be difficult [11], post-processing methods are not directly comparable with one another.

In this work, we present the first formal study of post-hoc methods for debiasing neural networks. A post-hoc method is defined as an algorithm which has access to a trained model and a validation dataset, and either fine-tunes the model or performs post-processing on the model predictions. We start by showing that the difficulty of post-hoc debiasing is highly dependent on the initial conditions of the original model. In particular, given a neural network trained to optimize accuracy, the variance in the amount of bias in the trained model is much higher than the variance in the accuracy, with respect to the random seed used for initializing the weights of the original model. Therefore, even the initial random seed can substantially change the outcome of post-hoc debiasing algorithms.

Next, we present three new optimization-based techniques for post-hoc debiasing of neural networks, each of which work for any group fairness measure. For each technique, we choose an objective upfront, which is a function of a fairness measure and model accuracy. We define a simple algorithm, random perturbation, which iteratively adds multiplicative noise to the weights of the neural network and then thresholds the output probabilities to minimize the objective function. Our second new technique is a layer-wise optimization algorithm. In this approach, we iteratively choose a layer of the neural network and use gradient-boosted regression trees to optimize the weights of the chosen layer with respect to the objective function. Our last technique is an adversarial fine-tuning algorithm. Adversarial trainnig is a powerful debiasing method because training a critic (discriminator) to predict bias effectively makes the objective function differentiable, enabling the use of first-order optimization techniques such as gradient descent. This has recently been proposed as an in-processing method for debiasing [50]. We show that using an adversarial model to fine-tune the trained neural network is a viable post-hoc technique.

We compare the three above techniques with three post-processing algorithms from prior work: reject option classification [27], equalized odds post-processing [24], and calibrated equalized odds post-processing [44]. We run experiments with three popular fairness datasets and three popular fairness definitions. We show that certain algorithms are useful in certain scenarios. For example, the random perturbation algorithm is a strong post-hoc debiasing baseline. The adversarial fine-tuning method is more powerful for debiasing larger models, but it is more computationally intensive and may require hyperparameter tuning. The layer-wise fine-tuning algorithm may work well on models in which the bias is concentrated in one layer.

Fairness research (and machine learning research as a whole) has seen a huge increase in popularity, and recent papers have highlighted the need for fair and reproducible results [47, 4]. To facilitate best practices, we run our experiments on the AIF360 toolkit [4] and open source all of our code.

**Our contributions.**    We summarize our main contributions below.

- We study the nature of post-hoc techniques for debiasing neural networks, showing that the problem is sensitive to the initial conditions of the original model.

- We present the first three measure agnostic, fine-tuning algorithms for post-hoc debiasing: random perturbation, layer-wise optimization, and adversarial fine-tuning. Our algorithms outperform all existing post-processing techniques on average.

- We conduct a study of post-hoc techniques for debiasing neural networks, testing six different algorithms across three datasets and with three different fairness measures.

## 2   Related Work

**Debiasing algorithms.**    There is a surging body of research on bias and fairness in machine learning. There are dozens of types of bias that can arise [34], and dozens of formal definitions of fairness

have been proposed [38]. Popular definitions include statistical parity/demographic parity [16, 30], equal opportunity (a subset of equalized odds) [23], and average absolute odds [4]. Many bias mitigation techniques have been proposed, which generally fall into three categories: pre-processing, in-processing, and post-processsing. Post-processing debiasing techniques are performed on a pretrained model and do not require access to the full training set. Therefore, these techniques are useful in a variety of settings in which retraining is costly or impossible due to computational costs or data limitations. All prior work on post-processing techniques use label-flipping methods such as randomly flipping labels until the true/false negative rates are equal, or flipping labels in a critical region of predicted probabilities near 0.5 [23, 44, 27]. Currently, these techniques have only been established for specific fairness measures. For a full overview, see [4, 49].

There are several pre- or in-processing optimization-based techniques for fairness. Prior work has used hyperparameter optimization to select parameters for training models to exhibit less bias [10]. This approach repeatedly retrains the full model with different hyperparameters, making it impractical for big data applications. Other work uses global optimization theory to build regression models that maximize accuracy and minimize correlation of the output with sensitive attributes [29]. Bias reduction has also been framed as a pre-processing convex optimization problem [8]. So far, these techniques have only been developed for specific fairness definitions. AdaFair [25] is a modification of AdaBoost [19] that updates the weights of training instances based on a bias measure. Another work uses a variant of Lagrangian multipliers to train a model with fairness constraints [13]. Prior work has also used adversarial neural network approaches to debias algorithms [50]. To the best of our knowledge, there are no post-hoc adversarial debiasing techniques.

## 3 Preliminaries

In this section, we give notation and definitions used throughout the paper. Given a dataset split into three parts, $\mathcal{D} = \mathcal{D}_{\text{train}} \cup \mathcal{D}_{\text{valid}} \cup \mathcal{D}_{\text{test}}$, let $(\boldsymbol{x}_i, Y_i) \in \mathcal{D}$ denote one datapoint, where $\boldsymbol{x}_i \in \mathbb{R}^d$ contains $d$ features including one binary protected attribute $A$ (e.g., identifying as female or not identifying as female), and $Y_i \in \{0, 1\}$ is the label. We denote a trained neural network by a function $f_\theta : \mathbb{R}^d \to [0, 1]$, where $\theta$ denotes the trained weights. We often denote $f_\theta(\boldsymbol{x}_i) = \hat{Y}_i$, the output predicted probability for datapoint $\boldsymbol{x}_i$. We denote a set of labels in $\mathcal{D}$ by $\mathcal{Y}$.

**Fairness measures.** We now give an overview of group fairness measures used in this work. Given a dataset $\mathcal{D}$ and protected attribute $A$, we define the true positive and false positive rates as

$$TPR_{A=a} = P(\hat{Y} = 1 \mid A = a, Y = 1), \text{ and } FPR_{A=a} = P(\hat{Y} = 1 \mid A = a, Y = 0),$$

where the probability is over $Y$ randomly drawn from the set of labels in a dataset $\mathcal{D}$.

*Statistical Parity Difference (SPD)*, or demographic parity difference [16, 30], measures the difference in the probability of a positive outcome between the protected and unprotected groups. Formally,

$$SPD(\mathcal{Y}, \hat{\mathcal{Y}}, A) = P_{\hat{Y} \in \hat{\mathcal{Y}}}(\hat{Y} = 1 \mid A = 0) - P_{\hat{Y} \in \hat{\mathcal{Y}}}(\hat{Y} = 1 \mid A = 1).$$

*Equal opportunity difference (EOD)* [23] measures the difference in TPR for the protected and unprotected groups. Equal opportunity is identical to *equalized odds* in the case where the protected feature and labels are binary. Formally, we have

$$EOD(\mathcal{Y}, \hat{\mathcal{Y}}, A) = TPR_{A=0} - TPR_{A=1}.$$

*Average Odds Difference (AOD)* [4] is defined as the average of the difference in the false positive rates and true positive rates for unprivileged and privileged groups. Formally,

$$AOD(\mathcal{Y}, \hat{\mathcal{Y}}, A) = \frac{(FPR_{A=0} - FPR_{A=1}) + (TPR_{A=0} - TPR_{A=1})}{2}.$$

**Optimization techniques.** Zeroth order (non-differentiable) optimization is used when the objective function is not differentiable (as is the case for most definitions of group fairness). This is also called black-box optimization. Given an input space $W$ and an objective function $\mu$, zeroth

order optimization seeks to compute $w^* = \arg\min_{w \in W} \mu(w)$. Leading methods for zeroth order optimization when function queries are expensive (such as optimizing a deep network) include gradient-boosted regression trees (GBRT) [20, 33] and Bayesian optimization (BO) [45, 18, 48], however BO struggles with high-dimensional data.

In contrast, first-order optimization is used when it is possible to take the derivative of the objective function. For example, gradient descent is a first-order optimization technique.

# 4  Methodology

In this section, we describe three new fine-tuning techniques for debiasing neural networks. First we give more notation and formally define the different types of debiasing algorithms.

Given a neural network $f_\theta$, we sometimes drop the subscript $\theta$ when it is clear from context. We denote the last layer of $f$ by $f^{(\ell)}$, and we assume that $f = f^{(\ell)} \circ f'$, where $f'$ is all but the last layer of the neural network. Our layer-wise optimization algorithm assumes that $f$ is feed-forward, that is, $f = f^{(\ell)} \circ \cdots \circ f^{(1)}$ for functions $f^{(1)}, f^{(2)}, \ldots, f^{(\ell)}$. The performance of the model is given by a performance measure $\rho$. For a set of points $\mathcal{D}' \subseteq \mathcal{D}$, given the set of true labels $\mathcal{Y}$ and the set of predicted labels $\hat{\mathcal{Y}} = \{f(\boldsymbol{x}_i) \mid (\boldsymbol{x}_i, Y_i) \in \mathcal{D}'\}$, the performance is $\rho(\mathcal{Y}, \hat{\mathcal{Y}}) \in [0, 1]$. Common performance measures include accuracy, precision, recall, or AUC ROC (area under the ROC curve). We also define a bias measure $\mu$, given as $\mu(\mathcal{Y}, \hat{\mathcal{Y}}, A) \in [0, 1]$, such as one defined in Section 3.

The goal of any debiasing algorithm is to minimize the bias $\mu$, without sacrificing performance $\rho$ too much. Many prior works have observed that fairness comes at the price of accuracy for many datasets, even when using large models such as deep networks [4, 49, 11], which means it is often not possible to achieve zero bias without significantly lowering accuracy. Therefore, a common technique is to minimize an objective function such as the following.

$$\phi_{\mu,\rho}(\mathcal{Y}, \hat{\mathcal{Y}}, A) = \lambda \cdot |\mu(\mathcal{Y}, \hat{\mathcal{Y}}, A)| + (1 - \lambda)(1 - \rho(\mathcal{Y}, \hat{\mathcal{Y}})). \tag{1}$$

In the expression, $\lambda$ is a parameter in $[0, 1]$ which can be tuned based on the desired bias or based on the level of bias in the original model.

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

5:     Select threshold $\tau \in [0, 1]$ which minimizes objective $\phi_{\mu,\rho}$
6:     Set val $= \phi_{\mu,\rho}(\mathcal{Y}, \{\mathbb{I}\{f_{\theta'}(\boldsymbol{x}) > \tau\} \mid (\boldsymbol{x}, Y) \in \mathcal{D}_{\text{valid}}\}, A)$, where $\theta' = \{\theta_1, ..., \theta'_i, ..., \theta_\ell\}$
7:     If val $<$ val$^*$ set val$^* = $ val, and $\theta^* = \theta'$.
8: **end for**
9: **Output:** $\theta^*, \tau^*$
---

**Adversarial fine-tuning.** The previous two methods rely on zeroth order optimization techniques because most group fairness measures such as statistical parity difference and equalized odds are non-differentiable. Our last technique casts the problem of debiasing as first-order optimization by using adversarial learning. The idea behind the adversarial method is that we train a critic model to predict the amount of bias in a minibatch. We sample the datapoints in a minibatch randomly and with replacement. This statistical bootstrapping approach to creating a minibatch means that if the critic can predict the bias in a minibatch accurately, then it can predict the bias in the model with respect to the validation set reasonably well. Therefore, the critic effectively acts as a differentiable proxy for bias, which makes it possible to debias the original model using gradient descent.

The adversarial algorithm works by alternately iterating between training the critic model $g$ using the predictions from $f$, and fine-tuning the predictive model $f$ with respect to $\phi_{\mu,\rho}$ using the bias proxy $\hat{\mu}$ from $g$. Note that the first layer in $g$ concatenates the minibatch and returns a single number that estimates the bias of the minibatch as the final output. See Algorithm 3.

---
**Algorithm 3** Adversarial Fine-Tuning
---
1: **Input:** Trained model $f = f_\ell \circ f'$ with weights $\theta$, validation dataset $\mathcal{D}_{\text{valid}}$, objective $\phi_{\mu,\rho}$ parameters $\lambda$, $m$, $m'$
2: Set $g$ as the critic model with weights $\theta'$.
3: **for** $i = 0$ to $n$ **do**
4:     **for** $j = 0$ to $m$ **do**
5:         Sample a minibatch $(\boldsymbol{X}_k, \boldsymbol{\mathcal{Y}}_k)$ with replacement from $\mathcal{D}_{\text{valid}}$
6:         Evaluate the bias in the minibatch, $\hat{\mu} \leftarrow \mu(\boldsymbol{\mathcal{Y}}_k, f(\boldsymbol{X}_k))$.
7:         Update the critic model $g$ by updating its stochastic gradient

$$\nabla_{\theta'}(\hat{\mu} - (g \circ f')(\boldsymbol{X}_k))^2$$

8:     **end for**
9:     **for** $j = 0$ to $m'$ **do**
10:        Sample a minibatch $(\boldsymbol{X}_k, \boldsymbol{\mathcal{Y}}_k)$ with replacement from $\mathcal{D}_{\text{valid}}$
11:        Update the original model by updating its stochastic gradient

$$\nabla_{\theta}\left[(1 - \lambda) \cdot (g \circ f')(\boldsymbol{X}_k) + \lambda \cdot \text{BCELoss}(\boldsymbol{\mathcal{Y}}_k, f(\boldsymbol{X}_k))\right]$$

12:     **end for**
13:     Select threshold $\tau \in [0, 1]$ that minimizes the objective $\phi_{\mu,\rho}$
14: **end for**
15: **Output:** Debiased model $f$, threshold $\tau$
---

## 5 Experiments

In this section, we experimentally evaluate the techniques laid out in Section 4 compared to baselines, on three datasets and with multiple fairness measures. To promote reproducibility, we use popular datasets from the AIF360 toolkit [4], and we release our code. Each dataset contains one or more binary protected features(s) and a binary label. We briefly describe them below.

The COMPAS dataset is a commonly used dataset in fairness research, consisting of over 10,000 defendants with 402 features [17]. The goal is to predict the recidivism likelihood for an individual [1]. We run separate experiments using *race* and also *sex* as protected attributes. The Adult Census Income (ACI) dataset is a binary classification dataset from the 1994 USA Census bureau database in which the goal is to predict whether a person earns above $50,000 [15]. There are over 40,000 data points with 15 features. We use *sex* as the protected attribute. The Bank Marketing (BM) dataset is from the phone marketing campaign of a Portuguese bank. There are over 48,000 datapoints consisting of 17 categorical and quantitative features. The goal is to predict whether a customer will subscribe to a product [35]. We use *age* as the protected feature.

**The need for neural networks.** First, we run a quick experiment to demonstrate the need for neural networks on the above datasets. Deep learning has become a very popular approach in the field of machine learning [32], however, for tabular datasets with fewer than 20 features, it is worth checking whether logistic regression or random forest techniques perform as well as neural networks [37]. We construct a neural network with 10 fully-connected layers, BatchNorm for regularization, and a dropout rate of 0.2, and we compare this to logistic regression and a random forest model on the ACI dataset. We see that a neural network achieves accuracy and area under the receiver operating characteristic curve (AUC ROC) scores which are 2% higher than the other models. See Appendix B for the full results. Therefore, for the rest of this section, we focus on using neural networks.

**Bias sensitivity to initial model conditions.** Next, we run experiments to compute the amount of variance in the bias scores of the initial models. Neural networks have a huge number of local minima. Hyperparameters such as the optimizer and learning rate, and even the initial random seed, cause the model to converge to different local minima [32]. Techniques such as the Adam optimizer and early stopping with patience have been designed to allow neural networks to consistently reach local minima with high accuracies [28, 21]. However, there is no guarantee on the amount of bias. In particular, the local minima found by neural networks may have large differences in the amount of bias, and therefore, there may be very high variance on the amount of bias exhibited by neural

Table 1: Bias and accuracy of a neural network.

| | AOD | EOD | SPD | accuracy |
|---|---|---|---|---|
| ACI (sex) | -0.084 ± 0.012 | -0.082 ± 0.017 | -0.198 ± 0.011 | 0.855±0.002 |
| BM (age) | 0.011 ± 0.027 | -0.009 ± 0.051 | 0.047 ± 0.015 | 0.901±0.002 |
| COMPAS (race) | 0.138 ± 0.017 | 0.194 ± 0.027 | 0.168 ± 0.016 | 0.669±0.006 |

networks just because of the random seed. Every local optima has a different set of weights. If the weights of the model at a specific local optimum rely heavily on the protected feature, removing the bias from such a model by updating the weights is harder than removing the bias from a model whose weights do not rely on the protected feature as heavily. Table 1 shows the mean and the standard deviation of three fairness measures, as well as accuracy, for training a neural network with 10 different initial random seeds, across three datasets. We see that the standard deviation of the bias score is an order of magnitude higher than the standard deviation of the accuracy. In Appendix B, we plot the contribution of each individual weight to the bias score, for a neural network. We show that the contribution of the weights to the bias score are sensitive to the initial random seed.

## 5.1 Post-hoc debiasing experiments

Now we present our main experimental study by comparing our three post-hoc debiasing methods to three baseline methods on three datasets and with three fairness measures. Note that we do not compare to any in-processing debiasing algorithms, because these algorithms require the entire training set, yet all post-hoc methods only use the validation set. We briefly describe the baseline post-processing algorithms that we tested.

The *reject option classification* post-processing algorithm [27] defines a critical region of points in the protected group whose predicted probability is near $0.5$, and flips these labels. This algorithm is designed to minimize statistical parity difference. The *equalized odds* post-processing algorithm [23] defines a convex hull based on the bias rates of different groups, and then flips the label of data points that fall inside the convex hull. This algorithm is designed to minimize equal opportunity difference. The *Calibrated equalized odds* post-processing algorithm [44] defines a base rate of bias for each group, and then adds randomness based on the group into the classifier until the bias rates converge. This algorithm is designed to minimize equal opportunity difference. For all algorithms, we use the implementations in the AIF360 repository [4].

Our initial model consists of a feed-forward neural network with 10 fully-connected layers of size 32, with a BatchNorm layer between each fully-connected layer, and a dropout fraction of 0.2. The model is trained with the Adam optimizer and an early-stopping patience of 100 epochs. The loss function is the binary cross-entropy loss. We use the validation data as the input for the post-hoc debiasing methods. The three post-hoc methods are set to optimize Equation 1 with $\lambda = 0.75$. We run each post-hoc method on 10 neural networks initialized with different random seeds. We present the results in Figure 1. Note that since the three post-processing baselines are only set up to minimize a specific fairness measure, there is only a fair comparison on their respective measures.

**Discussion.** We see that the three fine-tuning methods significantly outperform the baseline methods, sometimes even on the fairness metric for which the baseline was designed. We note that there are two caveats. First, the three fine-tuning methods had access to the objective function in Equation 1, while the post-processing methods are only designed to minimize their respective fairness measures. However, sometimes the fine-tuning algorithms are Pareto optimal compared to the baselines, with respect to accuracy and bias, as seen in the Pareto plots in Figure 1. Second, fine-tuning methods are more powerful than post-processing methods, since post-processing methods do not modify the weights of the original model, although it comes at the price of computation time (See Table 2). Post-processing methods are more appropriate when the model weights are unavailable or when computation time is constrained, and fine-tuning methods are more appropriate when higher performance is desired. We see that random perturbation is a very strong fine-tuning technique, performing the best in nearly every setting. Layer-wise optimization performs well in most settings, but is sometimes susceptible to the initial conditions of the original model which makes intuitive sense given the discussion earlier in this section on bias sensitivity to initial model conditions. There

Table 2: Runtime for every post-hoc algorithm for every dataset in seconds

|  | ACI (sex) | BM (age) | COMPAS (sex) | COMPAS (race) |
|---|---|---|---|---|
| ROC | 29.836 | 20.637 | 9.979 | 10.532 |
| EqOdds | 0.015 | 0.012 | 0.011 | 0.011 |
| CalibEqOdds | 0.144 | 0.064 | 0.049 | 0.054 |
| Random | 156.848 | 113.529 | 61.937 | 63.540 |
| Adversarial | 32.889 | 36.128 | 36.156 | 34.432 |
| LayerwiseOpt | 186.480 | 146.760 | 79.800 | 79.800 |

Figure 1: Results for post-hoc techniques. A lower objective score is better.

is only one setting in which adversarial fine-tuning outperformed random perturbation. This is likely due to the fact that adversarial fine-tuning is the most complex technique (training a neural network as a subroutine), but most of our datasets have at most 20 features. We hypothesize that adversarial fine-tuning will outperform random perturbation on more complex datasets, such as debiasing image datasets. In Appendix B, we run experiments with different types of initial neural networks.

## 6   Conclusion

In this work, we present the first study on post-hoc methods for debiasing neural networks. We present the first three measure-agnostic fine-tuning algorithms for debiasing neural networks: random perturbation, adversarial fine-tuning, and layer-wise optimization. First we show that the problem of post-hoc debiasing is sensitive to the initial conditions of the original neural network. Then we give an extensive study of post-hoc debiasing of neural networks by comparing our three new algorithms with three baseline post-processing algorithms on three popular fairness datasets and with three popular fairness measures. We show that each fine-tuning algorithm performs well for different datasets and different fairness metrics.

# 7 Broader Impact

Deep learning algorithms are becoming more prevalent than ever before. The technology is becoming more and more integrated into society. There are countless examples of machine learning, from recommender systems (every time you browse Netflix, Amazon, YouTube, Facebook, etc.) to self-driving cars, to life-impacting events such as criminal recidivism, loan repayment, and hiring decisions. It is also becoming increasingly more evident that all of these algorithms are biased from various sources [43, 40, 41]. Using technology for life-changing events which makes racist, sexist, and prejudiced decisions will only deepen the divides that exist in society. In the current social climate in countries such as the United States, this is not acceptable.

Our work seeks to decrease the negative effects that biased deep learning algorithms have on society. Our post-hoc methods, which work for any group fairness measure, will be immediately applicable to even large deep learning frameworks, since the models need not be retrained from scratch. Furthermore, we present simple techniques (random perturbation) as well as more complex and strong techniques (adversarial fine-tuning). For these reasons, we believe that our work has the potential to have an immediate impact in mitigating the bias in society. Furthermore, since we study the nature of post-hoc debiasing and present a study comparing prior work to our algorithms, our work may facilitate future work in post-hoc debiasing techniques.

## 7.1 Impact on bias in judicial applications

We briefly focus on how post-hoc methods for debiasing could help in judicial contexts. While, unfortunately, there may be innate human rights and legal issues in using machine learning applications in criminal justice contexts [46, 22], such applications are now widely deployed in many jurisdictions. Studies and investigations such as [31] have found that many of the algorithms used in judicial contexts have some form of bias. Moreover, different entities using the same model might prefer to use different fairness measures and some of these measures might be incompatible [12]. Glencora et al. [7] show that even social media monitoring tools used by law enforcement might be racially biased.

Generally, the entities that build and use the applications are not the same. Therefore, due to legal and licensing issues, the entity using the application may not have access to the training dataset for the model. This precludes the use of pre-processing and in-processing methods for debiasing. The entity using the model usually has its own dataset available (e.g. a local court tracking their recidivism rates). This makes post-hoc processing the only viable method for debiasing.

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

# A   Related Work Continued

In this section, we give a more detailed description of the related work from Section 2.

**Debiasing overview.**   There is a surging body of research on bias and fairness in machine learning. There are dozens of types of bias that can arise [34], and dozens of formal definitions of fairness have been proposed [38]. Popular definitions include statistical parity/demographic parity [16, 30], equal opportunity (a subset of equalized odds) [23], and average absolute odds [4]. Many bias mitigation techniques have been proposed, which generally fall into three categories: pre-processing, in-processing, and post-processsing. Post-processing debiasing techniques are performed on a pretrained model and do not require access to the full training set. Therefore, these techniques are useful in a variety of settings in which retraining is costly or impossible due to computational costs or data limitations.

**Post-processing methods**   All prior work on post-processing techniques use label-flipping methods such as randomly flipping labels until the true/false negative rates are equal, or flipping labels in a critical region of predicted probabilities near 0.5 [23, 44, 27]. Currently, these techniques have only been established for specific fairness measures. For a full overview, see [4, 49].

We gave brief descriptions of three post-processing debiasing techniques in Section 5, which we restate here for convenience. In reject option classification, a critical region of points in the protected group is defined, whose predicted probability is near $0.5$, and these labels are then flipped [27]. The equalized odds post-processing algorithm defines a convex hull based on the false and true positive rates of different groups, and then flip the label of data points that fall into the convex hull [23]. In the calibrated equalized odds post-processing algorithm, a base rate of false negatives is defined for each group, and then randomness is added based on the group into the classifier until the false negative rates are equal [44]. Currently, these techniques have only been established for specific fairness measures.

**Hyperparameter optimization for fairness**   There is a wide variety of work on in-processing debiasing algorithms which are similar in spirit to our optimization methods. We mention a few of them here. However, none of them explicitly present a post-hoc debiasing algorithm. Recently, a meta-algorithm was developed for in-processing debiasing by reducing many fairness measures to convex problems [9]. Another work treats debiasing as an empirical risk minimization problem [14]. Yet another work adds the fairness constraints as regularizers in the machine learning models [5]. Other prior work has used hyperparameter optimization to select parameters for training models to exhibit less bias [10]. This approach repeatedly retrains the full model with different hyperparameters, making it impractical for big data applications. Other work uses global optimization theory to build regression models that maximize accuracy and minimize correlation of the output with sensitive attributes [29]. Bias reduction has also been framed as a pre-processing convex optimization problem [8]. The last two techniques have only been developed for specific fairness definitions.

AdaFair [25] is a modification of AdaBoost [19] that updates the weights of training instances based on a bias measure. Another work uses a variant of Lagrangian multipliers to train a model with fairness constraints [13]. Prior work has also used adversarial neural network approaches to debias algorithms [50]. To the best of our knowledge, there are no post-hoc adversarial debiasing techniques.

# B   Additional Experiments and Details

In this section, we give additional details from the experiments in Section 5, as well as additional experiments.

**The need for neural networks.**   We start by comparing the performance of neural networks to logistic regression and gradient-boosted regression trees (GBRT) on the datasets we used, to demonstrate the need for neural networks. This experiment is described at the start of Section 5. For convenience, we restate the details here. We construct a neural network with 10 fully-connected layers of size 32, BatchNorm for regularization, and a dropout rate of 0.2, and we compare this to logistic regression and GBRT on the ACI, BM, and COMPAS datasets. See Table 3. We see that the

Table 3: Comparison between models. mean $\pm$ standard deviation

|  |  | logistic regression | neural network | random forest |
|---|---|---|---|---|
| ACI | accuracy | $0.852 \pm 0.000$ | $\mathbf{0.855 \pm 0.002}$ | $0.844 \pm 0.002$ |
|  | roc_auc | $0.904 \pm 0.000$ | $\mathbf{0.908 \pm 0.001}$ | $0.889 \pm 0.000$ |
| BM | accuracy | $\mathbf{0.901 \pm 0.000}$ | $\mathbf{0.901 \pm 0.002}$ | $0.899 \pm 0.001$ |
|  | roc_auc | $0.930 \pm 0.000$ | $\mathbf{0.934 \pm 0.001}$ | $0.932 \pm 0.001$ |
| COMPAS | accuracy | $\mathbf{0.677 \pm 0.000}$ | $0.641 \pm 0.061$ | $0.652 \pm 0.006$ |
|  | roc_auc | $\mathbf{0.725 \pm 0.000}$ | $0.679 \pm 0.088$ | $0.695 \pm 0.002$ |

Figure 2: Results for coefficient analysis.

neural network achieves better accuracy and ROC AUC on all datasets except COMPAS, which is within one standard deviation of the optimal performance.

**Bias sensitivity to initial model conditions.** Next, we study the sensitivity of bias to initial model conditions. Recall that in Table 1, we computed the mean and standard deviation of three fairness measures, as well as accuracy, for training a neural network with respect to different initial random seeds. We see that standard deviation of the bias is an order of magnitude higher than the standard deviation of the accuracy. Now we run more experiments to show that the contribution of the weights to the bias score are sensitive to the initial random seed.

For this experiment, we trained 10 neural networks with the same architecture as described in the main body of the paper. We wanted to identify which parameters of the network contributed most to the bias. To identify these parameters, we created 1000 random delta vectors with mean 1 and standard deviation 0.1 for each of the neural networks. We then took the Hadamard product of each random delta vector with the parameters of the corresponding network. We then evaluated the bias (SPD) on the test set for the networks with the new perturbed parameters. To identify which parameters contributed most to the bias, we trained a linear model for each of the 10 neural networks to predict the bias from the random delta vectors and analyzed the coefficients of the corresponding linear models. The linear models were successfully able to predict the bias based on the random delta vectors with an $R^2$ score of $0.861 \pm 0.090$. Figure-2 (left) shows that only a small fraction of the parameters contribute to the majority of the bias.

Now we want to identify how similar the coefficients of the linear models are across all 10 neural networks. To identify this we stacked the normalized coefficients for the linear models and decomposed the stacked matrix with singular value decomposition. The singular values of the matrix measured the degree of linear independence between the coefficients for the 10 linear models. As we see from Figure-2 (right) the singular values are all close to 1. This indicates that the coefficients are relatively different from each other. This means that the parameters of the 10 neural networks that correspond to the bias are different for each network indicating that each time we train a model – even if it has the same architecture – the parameters that contribute to bias are different.

**Additional debiasing experiments** To show the robustness of our fine-tuning methods to variations in model architecture we run a suite of debiasing experiments on the Adult Census Income dataset. We train four variations of our initial model and run all debiasing experiments on all four variations. For the first variation we trained a model with dropout rate = 0.3, hidden layer width = 32, and number of stacked fully connected layers = 2. For the second variation we trained a model with

Figure 3: Results from running variations of neural network architecture.

dropout rate = 0.5, hidden layer width = 16, and number of stacked fully connected layers = 10. For
the third variation we trained a model with dropout rate = 0.5, hidden layer width = 128, and number
of stacked fully connected layers = 2. For the fourth variation we trained a model with dropout rate =
0.2, hidden layer width = 64, and number of stacked fully connected layers = 15. We run 10 trials of
each neural network, which involves training the neural network with a new random seed, and then
applying all post-hoc debiasing methods. We average all 40 trials across the 4 neural networks, and
give the results in Figure 3.

[Supplementary Material 2 · spd_results.pdf]

**Statistical Parity Difference**

Legend: Default, ROC, EqOdds, CalibEqOdds, Random, Adversarial, LayerwiseOpt

X-axis (Dataset): ACI (sex), BM (age), COMPAS (sex), COMPAS (race)

Y-axis (Objective): 0.0, 0.1, 0.2, 0.3, 0.4

[Supplementary Material 3 · eod_results.pdf]

**Equal Opportunity Difference**

Legend: Default, ROC, EqOdds, CalibEqOdds, Random, Adversarial, LayerwiseOpt

X-axis (Dataset): ACI (sex), BM (age), COMPAS (sex), COMPAS (race)

Y-axis (Objective): 0.0 to 0.7

[Supplementary Material 4]

Pareto Plot ACI (sex)

Legend: Default, ROC, EqOdds, CalibEqOdds, Random, Adversarial, LayerwiseOpt