[Reviews · NeurIPS 2020]

Review 1

Summary and Contributions: The authors compared three approaches to debias a neural network without a need to retrain the whole model.

Strengths: The proposed debiasing algorithms are inspired by hyper-parameter tuning in machine learning. They propose simple and fast to more complex and time consuming methods. It is nice that their approach does not depend on a single fairness notion and can work with many types of fairness notions.

Weaknesses: The novelty of their approaches is questionable. They performed the unbiasing task similar to parameter tuning, however treating discrimination by just adding a parameter to the objective won't solve the issue, and there is no guarantee that the outcome is fair. Authors only consider the case for one sensitive attribute and they didn't mention how their approach may extend to multiple attributes. Considering more attributes is more realistic since even for one the datasets that they tried, i.e., COMPAS, they had to try their model independently for two different attributes, i.e., sex and race, and this is not feasible to debias them both at the same time.

Correctness: To evaluate their approach, they only report accuracy and three fairness measures on three datasets (and for COMPAS they didn't add it for sex) before debiasing, and there is no report for the same values after debiasing with their approaches, which is not related to the main contribution of the paper. From the experimental setup, the advantage of their complex model is not clear. They are more time consuming and the results are not necessarily better than related work or their simple random model.

Clarity: The paper is easy to read and clear but the figures are not easy to read and tiny, I suggest to the authors to either the size or at-least increase the font size on these figures.

Relation to Prior Work: The authors only compare with post-processing approaches that work with one specific measure. However, they can only compare their random approach with those approaches, since it does not need to have access to the original model. Regarding the other two approaches, while they do not re-train the whole model, they re-train part of the model and would be good if the authors could compare with other approaches that debias during processing.

Reproducibility: Yes

Additional Feedback: - What is the objective in figure 1? - Why you use 0.75 as the value of lambda for the experiment? - What are the value of the ACC, AOD, EOD, SPD after debiasing? ================================= EDIT AFTER AUTHOR RESPONSE ================================= I read the authors' response and thank them for answering the questions. I encourage the authors to include the experiments that they ran for the rebuttal in the updated version of the paper and compare their approach with other existing in-processing work on de-biasing NNs.


Review 2

Summary and Contributions: Debiasing methods could be categorized into pre-, in-, and post-processing algorithms. The authors of this paper provided 3 new post-processing/post-hoc algorithms for debiasing neural network models. They proposed fine-tuning methods that are different from existing methods as they modify the weights of the original model. Algorithm 1: Using the validation set, in every iteration the weights of the neural network will be perturbed by a Gaussian random variable to minimize the objective function specifically designed for the task of debiasing. Algorithm 2: Instead of using a random perturbation, an optimization algorithm will be used. The weights in each layer, using gradient-boosted regression trees, will be optimized with respect to the new objective function. Algorithm 3: A critic model is trained to predict the amount of bias in a minibatch. If the critic was successful in predicting the bias in a minibatch, then it can act as a differentiable proxy for the bias of the model. Finally, using gradient descent and the learned critic model, the gradients of the original model will be modified. These three new algorithms were compared with three other methods with respect to three common fairness measures.

Strengths: 1. The authors provided a detailed description of their three methods. The pseudo-code of their algorithms made it so easy to understand the steps. 2. The paper is so well-written and well-structured. The authors tried to answer every possible question that might arise. 3. Their methods are technically sound but lack some additional experiments. With a little bit of effort this paper can be improved and ready to publish.

Weaknesses: 1. It would be clearer if the accuracy rates of the methods were provided in a table. 2. The authors claimed that the method introduced in the paper “Mitigating Unwanted Biases with Adversarial Learning” is an in-processing algorithm. While this is not true as this method is a model-agnostic post-processing method that only changes the weights of the adversary. So, this model should be studied in the experiments. 3. The weight of the original model in all three methods should be known beforehand, while most of the cases we are dealing with black boxes. 4. Similar papers in the area are not mentioned, such as “Multiaccuracy: Black-Box Post-Processing for Fairness in Classification”, “Towards Debiasing Sentence Representations”, and “Man is to Computer Programmer as Woman is to Homemaker? Debiasing Word Embeddings”. ******************************************* After going through the rebuttal, even though the authers have presented more rexperimental results, I am still not clear about the motivation behind proposing algorithm 2 and 3 especially since the results show not much gain compared to the first algorithm. I also believe the authors still need to run more experiments and compare with more exiting works. Therefore, I will keep my score unchanged.

Correctness: Some claims are incomplete. For example the structure of the critic model in the third algorithm was not defined (Some other examples can be found in the “Weakness” section of this review).

Clarity: The paper is well-written and well-structured. Some questions were simply answered by the authors before even being raised.

Relation to Prior Work: Yes, it is.

Reproducibility: Yes

Additional Feedback:


Review 3

Summary and Contributions: The authors proposed post-hoc debiasing methods for deep neural networks. While most studies try to build fair models by incorporating specific regularizations/constraints during the model training, this study focuses on the post-hoc debiasing of the trained models. Post-hoc debiasing would be important when the one that trained the model and the one that uses the model are different, where the users cannot access the training data that are used for training the model. The authors proposed three techniques for post-hoc debiasing, and found that even the simple random search is still helpful in practice, which will be a standard baseline in the future researches.

Strengths: Tackling the post-hoc debiasing itself will be a novelty of this paper. The motivation, the demand of the post-hoc debiasing, is practical and convincing. Three methods considered in this paper are reasonable and are easy to implement.

Weaknesses: One weakness (that can be fixed) is that the size of datasets used in the experiments are missing in the paper. I am particularly curious how large the validation set should be so that the methods to work. Apparently, too small validation set will not be helpful. On the other hand, if the method requires large validation set, the users may be able to train a new neural network from scratch while incorporating fairness regularizations/constraints. It would be appreciated if there is an analysis on how the size of the validation set affects the result.

Correctness: I think there is no major flaw.

Clarity: The idea of the paper is easy to follow.

Relation to Prior Work: The authors discuss in detail how the current study differs from the prior studies.

Reproducibility: Yes

Additional Feedback: For the major comment, please refer to Weakness. I would like to suggest the authors to improve the visibility of Figure 1, e.g. by increasing the font size or by changing the colors. The current figure is hard to check. === update after rebuttal === After the discussion with other reviewers, I decided to decrease my score because the paper [Ref] raised by Reviewer2 seems to be an important prior work. Multiaccuracy Boost of [Ref] debiases the models using the validation set, i.e. [Ref] tackles essentially the same problem as the current paper. The paper will be much stronger by enriching the discussions and empirical evaluations comparing the proposed methods, such as random perturbation, with Multiaccuracy Boost. [Ref] “Multiaccuracy: Black-Box Post-Processing for Fairness in Classification”


Review 4

Summary and Contributions: In this paper, they introduced three post-hoc methods to debias neural networks by fine-tuning them: 1) random perturbation in which they add Gaussian noise to the weights, 2) layer-wise optimization in which they used zeroth-order optimization, and 3) adversarial fine-tuning in which we train a model to predict the bias. They performed their experiments on three dataset and compare these three debiasing method with respect to various fairness criterion.

Strengths: - The paper is well-written and well-organized. - They address a very important issue in machine learning. - Introduced methods are post-hoc, and could be applied when we do not have access to the model's parameters. Especially, their random perturbation, which is the most straightforward and most efficient among the rest, has high outperformance. - They discuss the computational costs of their propose methods and list them in Table 2.

Weaknesses: - I tried to understand why these three methods helping models to reduce the bias. It would be interesting to add a paragraph explaining the intuition behind the ideas and why these methods help for debiasing. - It would be interesting to see some qualitative examples based on these post-hoc debiasing. For example, showing one or two false positives that are correctly classified when applying debiasing. - They could perform more experiments using more datasets with respect to various targeted group (race, religion, gender).

Correctness: yes

Clarity: yes

Relation to Prior Work: yes

Reproducibility: Yes

Additional Feedback: I will keep my score the same, the addressed my two concerns, added qualitative analysis and add one more experiments. But did the explain the intuition behind their method.

[Author Response · NeurIPS 2020]

We thank all of the reviewers for their helpful reviews. We performed all of the experiments that were suggested: *(1)*
we changed the objective function so that a specified level of fairness is guaranteed; *(2)* we performed experiments
where there are multiple protected attributes simultaneously (e.g. race and gender on COMPAS); *(3)* we added a large
image dataset (CelebA) to our experiments, which shows that adversarial fine-tuning is the most powerful method for
complex tasks; *(4)* we implemented the algorithm from a similar paper, Zhang et al. "Mitigating Unwanted Biases with
Adversarial Learning", and we show that it does not outperform our methods. We hope that these experiments help to
address the reviewer concerns. Please see the details below.

**R1**  We agree that our original objective function has no guarantee that the outcome is fair. Therefore, we perform an
additional experiment where the objective function is $1 - \text{acc} \cdot \left(1 + e^{500(|\text{bias}|-0.03)}\right)^{-1}$, which enforces the constraint
$|\text{bias}| < 0.03$ since it behaves like a (smoothed) indicator function. See the table below (row 1). Next, we show that our
algorithms perform well when there are multiple protected attributes. We ran an experiment on COMPAS protecting
gender and race, as the reviewer suggested. See the table below (row 2). Next, we ran our experiments on a more
complex dataset (the image dataset CelebA), and this one clearly shows the effectiveness of adversarial fine-tuning over
the simpler methods (which was a concern of the reviewer). We used a ResNet trained to predict whether a person
is smiling, and we debias with respect to race. See the table (row 3) and the figure below. We agree that we should
compare our approach to other in-processing approaches (see our response to R2 for one of them). We also note that
comparing to in-processing algorithms is sometimes impossible. For example, the in-processing algorithms require the
full training dataset, while post-hoc methods only use the validation dataset. Furthermore, it is common to start with a
large, expensive model such as GPT-3 or EfficientNet, where retraining from scratch is infeasible. Finally, we agree
with all the smaller comments/clarifications (such as adding a table with accuracy/bias before and after debiasing) and
we will correct these in the final version of our paper.

**R2**  As suggested, we compared our approaches to Zhang et al. "Mitigating Unwanted Biases with Adversarial
Learning". Note there is a key difference: in that paper, the critic model learns to predict the protected attribute, while
in our paper, the critic directly predicts bias. See the table below (row 4 and column 8) for the results, which shows that
our methods outperform Zhang et al. Finally, we agree with all the smaller comments and will incorporate them into the
paper: adding the three papers to related work, including all details of the critic model, and adding a full table of our
results for clarity.

**R3**  We will make sure to include the size of the datasets in the final version of the paper, which are as follows (and we
use a train/val/test split of 60/20/20): ACI: $48842$, BM: $45211$, COMPAS: $10331$, and CelebA (see response to R1):
$60000$. We see generally that random search performs better on smaller datasets, and adversarial fine-tuning performs
better on large datasets. Finally, we will improve the visibility of Figure 1 in our paper.

**R4**  We will make sure to give much more intuition behind our methods in the final version of the paper. We agree that
we should use more datasets, so we ran experiments on the CelebA image dataset (see our response to R1). It is a great
idea to include qualitative examples. We give some examples below for the CelebA dataset.

|  |  | Default | ROC | EqOdds | CalibEqOdds | Random | Adversarial | LayerwiseOpt | Zhang et al. |
|---|---|---|---|---|---|---|---|---|---|
| (1) Bias Guarantee (on ACI) | objective (See R1) | 1 | 1 | 1 | 1 | 0.18 | 0.2 | 1 | 0.23 |
|  | \|bias\| | 0.082 | 0.055 | 0.15 | 0.31 | 0.02 | 0.011 | 0.14 | 0.006 |
|  | performance | 0.85 | 0.79 | 0.94 | 0.84 | 0.83 | 0.8 | 0.56 | 0.77 |
| (2) COMPAS (race & gender) | objective (Eq 1) | 0.23 | 0.14 | 0.15 | 0.33 | 0.15 | 0.21 | 0.14 | 0.16 |
|  | \|bias\| | 0.18 | 0.0016 | 0.17 | 0.18 | 0.025 | 0.14 | 0 | 0.052 |
|  | performance | 0.66 | 0.54 | 0.91 | 0.34 | 0.56 | 0.61 | 0.54 | 0.58 |
| (3) CelebA (race) | objective (Eq 1) | 0.059 | 0.072 | 0.25 | 0.086 | 0.051 | 0.046 | 0.25 | — |
|  | \|bias\| | 0.025 | 0.052 | 0.48 | 0.05 | 0.0077 | 0.0054 | 0 | — |
|  | performance | 0.91 | 0.91 | 0.98 | 0.88 | 0.91 | 0.91 | 0.5 | — |
| (4) Comp. w. Zhang et al. (on ACI) | objective (Eq 1) | 0.098 | 0.094 | 0.13 | 0.27 | 0.053 | 0.063 | 0.062 | 0.067 |
|  | \|bias\| | 0.082 | 0.055 | 0.15 | 0.31 | 0.0011 | 0.0047 | 0 | 0.025 |
|  | performance | 0.85 | 0.79 | 0.94 | 0.84 | 0.79 | 0.76 | 0.75 | 0.81 |

| | | | | | | | | | | | |
|---|---|---|---|---|---|---|---|---|---|---|---|
| Before | 74% | 28% | 23% | 51% | 77% | 5% | 3% | 93% | 99% | 2% | 99% | 83% |
| After | 92% | 35% | 29% | 61% | 84% | 11% | 2% | 92% | 99% | 2% | 99% | 84% |

Figure 1: Probability of smiling on the CelebA dataset, before and after debiasing w.r.t. race.

[Meta-Review · NeurIPS 2020]

This work presents a simple idea that works. The proposal can benefit from an improved presentation of the material and a more comprehensive related work section. In particular, prior work such as Multiaccuracy: Black-Box Post-Processing for Fairness in Classification should be cited. Besides, we strongly encourage the authors to incorporate in the final version of the paper the results presented in the rebuttal. Also, we suggest the authors present the results in a standard way that makes comparison with other work easier.